# TWIGMA: A dataset of AI-Generated Images with Metadata From Twitter

**Yiqun T. Chen**
Department of Biomedical Data Science
Stanford University
Stanford, CA 94305
yiqunc@stanford.edu

**James Zou**
Departments of Biomedical Data Science;
Electrical Engineering; and Computer Science
Stanford University
Stanford, CA 94305
jamesz@stanford.edu

## Abstract

Recent progress in generative artificial intelligence (gen-AI) has enabled the generation of photo-realistic and artistically-inspiring photos at a single click, catering to millions of users online. To explore how people use gen-AI models such as DALLE and StableDiffusion, it is critical to understand the themes, contents, and variations present in the AI-generated photos. In this work, we introduce TWIGMA (TWItter Generative-ai images with MetadatA), a comprehensive dataset encompassing over 800,000 gen-AI images collected from Jan 2021 to March 2023 on Twitter, with associated metadata (e.g., tweet text, creation date, number of likes), available at https://zenodo.org/records/8031785. Through a comparative analysis of TWIGMA with natural images and human artwork, we find that gen-AI images possess distinctive characteristics and exhibit, on average, lower variability when compared to their non-gen-AI counterparts. Additionally, we find that the similarity between a gen-AI image and natural images is inversely correlated with the number of likes. Finally, we observe a longitudinal shift in the themes of AI-generated images on Twitter, with users increasingly sharing artistically sophisticated content such as intricate human portraits, whereas their interest in simple subjects such as natural scenes and animals has decreased. Our findings underscore the significance of TWIGMA as a unique data resource for studying AI-generated images.

## 1 Introduction

Recent advancements in text-to-image generation models, such as DALLE [33] and StableDiffusion [37], have revolutionized the creation of realistic and visually captivating images. The ability to generate artistically inspiring images at the click of a button has attracted millions of daily active users. However, the surge in popularity has also given rise to intriguing questions about the boundaries of human and model creativity, as well as the diversity in topics and styles of the generated images. For instance, let's consider the StableDiffusion model: it passes the user-specified text input (known as *prompts*) into a text encoder CLIP [32] to obtain an encoding that captures the essence of the prompt. Next, the model utilizes a latent diffusion model that takes the text embedding as input and stochastically generates an image that is guided by the text encoding. Therefore, the output image is influenced by both the user's input prompt and the complex diffusion process. Consequently, the stochasticity and black-box nature of the diffusion process led to extensive practice and research in the field of prompt engineering, where content creators iteratively refine their text prompts to generate images that align closely with their desired targets [22, 28]. Similar to other emerging technologies with significant capabilities, generative image models have given rise to a plethora of intricate legal and ethical challenges. These include concerns such as the proliferation of AI-generated NSFW

37th Conference on Neural Information Processing Systems (NeurIPS 2023) Track on Datasets and Benchmarks.

(Not-Safe-For-Work, which broadly includes offensive, violent, and pornographic topics) images, the dissemination of fake news, and controversies surrounding copyright.

Given the ever-increasing popularity and controversy surrounding AI-generated images, understanding their themes, content, and variations is increasingly crucial. However, existing datasets available for research purposes do not sufficiently address these specific inquiries, as many of them were created for specialized investigations (e.g., fake art detection [44, 51]) and image quality evaluation [40]), resulting in limited content diversity. Some recent exceptions include the Kaggle dataset on Midjourney prompts [3] and DiffusionDB [52], two large-scale datasets compiled from Discord for Midjourney and StableDiffusion models, respectively. However, these pioneering datasets are limited in terms of model variations, user distribution, and relatively short data collection periods.

To address these limitations, we present TWIGMA (Twitter Generative-AI Images with MetadatA) — a large-scale dataset encompassing 800,000 AI-generated images from diverse models. Spanning January 2021 to March 2023, TWIGMA covered an extended timeframe and included valuable metadata, such as inferred image subjects and number of likes. To the best of our knowledge, TWIGMA is the *first* AI-generated image dataset with a substantial time span and rich metadata, enabling analysis of temporal trends in human-AI-generated image content. Moreover, we leverage unsupervised learning techniques and inferred image captions to understand themes of AI-generated images. This complements earlier research, which primarily concentrated on logged prompts' subjects and content [52, 53]. We demonstrate the value and the type of insights TWIGMA enables by using it to characterize the underlying themes and novel aspects of AI-generated images. In addition to offering opportunities for fine-tuned adaptations to generate images with high popularity on social platforms like Twitter, our dataset and accompanying analyses contribute substantial insights into the ways in which humans engage with generative models, a topic of much interest to a broader scientific framework: as the prevalence of generative content continues to grow, our study could serve as a pilot for future endeavors aimed at comprehending and enhancing the sociological [38] and human-machine interaction [9, 27] dimensions inherent to these data-driven gen-AI models.

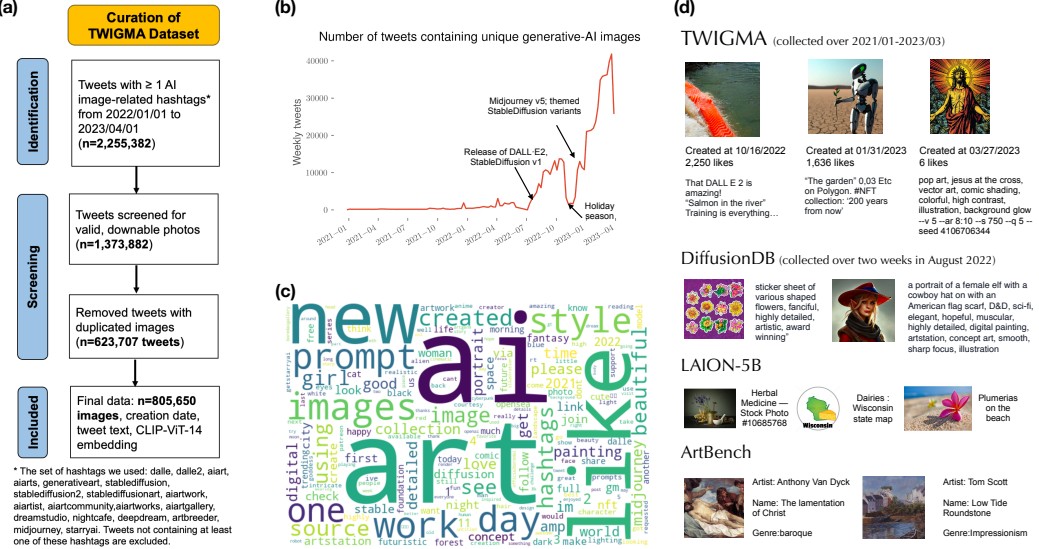

Figure 1: **Creation process and content overview of TWIGMA**. **(a):** Curation process of the TWIGMA dataset, resulting in approximately 800,000 images posted from 2021 to March 2023. **(b):** Steady growth in the count of tweets featuring generative AI images over time. **(c):** Wordcloud showcasing the prevalent keywords extracted from the tweets; popular words include new, AI, art, and prompts. **(d):** Schematic depiction of the data and metadata incorporated in our analyses.

The rest of the paper is organized as follows. We review image datasets and evaluation metrics for model novelty and variation in Section 2. The data collection and analysis processes are outlined in Section 3. Section 4 presents empirical results addressing our research questions. Finally, Section 5 discusses limitations, safety and ethics concerns, and future research directions.

Table 1: Comparing TWIGMA with other image datasets analyzed in the paper.

| Dataset | Source | # Images | # Prompts/Captions | Additional Features |
|---|---|---|---|---|
| **TWIGMA** (This paper) | Twitter Jan. 2020–Mar. 2023 | 800,000 | Inferred BLIP captions | Metadata (date, likes, clustering). Images come from multiple gen-AI models. |
| DiffusionDB [52] | Stable Diffusion Discord August 2022 | 18 Million | 1.8 unique prompts | Creation date, user hash |
| ArtBench[20] | WikiArt, Ukiyo-e.org 1400–2000 | 60,000 | Painting names | Artist names, creation time, art genre. |
| LAION-5B [42] | Common Crawl Filtered for image-text similarity | 5.85 Billion | Corresponding text | NSFW and watermark detection |

## 2 Related work

**AI-generated image dataset:** The creation of large-scale image-text datasets has rapidly evolved, transitioning from carefully annotated datasets relying on human labels [16, 21] to vast collections of image-text pairs gathered from the web [32, 37, 47]. As text-to-image models continue to demonstrate unprecedented capabilities in generating images based on user prompts, researchers have started curating similar datasets featuring images generated by these models. For example, DiffusionDB and Midjourney Kaggle provide large-scale datasets (14 million and 250,000, respectively) with prompts and images generated by StableDiffusion and Midjourney, respectively. However, these pioneering, general-purpose datasets are limited in terms of style (as they originate from a single model variation), user distribution (restricted to Discord users of specific channels), and relatively short data collection periods (one month in 2022 for both datasets). Researchers have also constructed datasets of AI-generated images for specialized use, such as detecting generated art images [46, 51], evaluating qualities of specific contents such as human portraits [6, 18], investigating safety filters and hidden vocabularies [26, 34], and evaluating potential biases [24]. Given the problem-oriented nature, these datasets are often limited in size and skewed in themes. Recognizing the gap between existing datasets and our research questions on novelty, themes, and variation of AI-generated images, we curated TWIGMA to demonstrate its potential in this paper.

**Novelty and variation of AI-generated images:** Many text-to-image models rely on continuous training with a substantial amount of human artistic artifacts sourced from datasets such as LAION [42], which includes images from platforms such as Wikiart and Pinterest, and potentially copyrighted or proprietary contents [31, 34, 44]. Users have also fine-tuned the open-sourced StableDiffusion model on additional samples from specific artists or art genres, such as anime, to generate AI-generated art imitating those styles [5, 8, 23]. While prior research has examined the reproduction of training data in AI-generated art settings [46, 51] and from the perspective of adversarial attack [7, 26], large-scale empirical results comparing AI- and non-AI-generated images are relatively limited. This prompts our first research question (RQ) to investigate the novelty of AI-generated images:

RQ1: How distinct are the distributions of AI-generated images from those of non-AI images?

Furthermore, the stochastic nature of many text-to-image models has sparked inquiries about the variability of their generated image outputs. In a notable court case, a judge determined that AI-generated images do not qualify for copyright protection, citing the significant disparity between the user's intended prompts for Midjourney and the resulting visual material it produced. Recent research has explored the variation within the prompt space [22, 52, 53], and our second RQ builds upon these findings and zooms in a particular dimension of RQ1, the *variation* in the image space:

RQ2: How do AI-generated image variations compare to non-AI-generated counterparts?

Lastly, we leverage the extended time span offered by TWIGMA to explore our final RQ:

RQ3: How do the content and theme of AI-generated images change over time?

By answering RQ1–3, our paper makes the following contributions to the literature: Firstly, we utilize metrics proposed in generative model evaluation to demonstrate the distinctiveness of AI-generated images compared to real images, providing evidence of the novelty embedded in the underlying models. Moreover, we find that the distance between AI-generated and human images (i) correlates with the number of likes received; and (ii) can be leveraged to identify human images that served as inspiration for AI-generated creations. Secondly, we introduce quantitative measures to establish

that (i) images from generative models exhibit less diversity than real images; and (ii) a substantial portion of variations in the image space can be attributed to the variation in user prompts. Finally, we observed a shift in the preferences for generative models and image topics among Twitter users: they are increasingly sharing artistically sophisticated or distinct content, such as intricate human portraits and anime, while interest in simpler subjects like natural scenes and animals has declined.

## 3  Methods

**Curating the TWIGMA dataset:**  To curate the TWIGMA dataset, we began with the initial filtering of available tweets using hashtags commonly associated with AI-generated images, such as `#dalle`, `#stablediffusion`, and `#aiart`. We then proceeded to iteratively refine the set of hashtags by incorporating new hashtags that co-occur with the existing ones and contain highly relevant tweets with AI-generated images. We evaluated the quality of these new hashtags based on the first 20 tweets returned by the hashtag search. This iterative process allowed us to create a final set of 19 hashtags, encompassing both generic community descriptions (e.g., `#aiart`, `#generativeart`) and specific models (e.g., `#midjourney`, `#dalle2`); please refer to Figure 1(a) for the complete list of hashtags.

Next, we utilized the official Twitter API to scrape tweets containing at least one of the identified hashtags. This process encompassed the time frame from January 1, 2021 to March 31, 2023, resulting in approximately 2.2 million tweets. Out of these potentially duplicated tweets, around 1.3 million contained downloadable photos as of April 2023. To ensure data quality, we conducted a two-step deduplication process. Firstly, we utilized the media ID in Twitter to retain only one image in cases where the same tweet was associated with multiple hashtags. Secondly, we computed CLIP-ViT-L-14 embeddings [32] to remove images with identical embeddings. This deduplication step resulted in a dataset of 623,707 tweets and 805,650 images in TWIGMA.

Furthermore, we extracted comprehensive metadata for each image in the TWIGMA dataset, including original tweet texts, engagement metrics such as likes, as well as user-related information such as follower counts. Recognizing that tweet texts may not always accurately describe the image content, we also generated captions for each image in TWIGMA using the BLIP model [19]. We release TWIGMA at `https://zenodo.org/records/8031785` and provide a tutorial at `https://yiqunchen.github.io/TWIGMA/`. This dataset includes our curated metadata pertaining to the image creation date, number of likes, assigned cluster membership obtained through k-means clustering, and the inferred BLIP captions. Additionally, we provide a list of Twitter IDs and the necessary code to retrieve images and metadata using the Public Twitter API.

This paper presents a series of analyses to showcase the fascinating insights that TWIGMA can offer into human-AI-generated images. Our approaches represent an initial exploration effort, and numerous other intriguing questions can be addressed using TWIGMA.

**Measuring distinctiveness of AI-generated images:**  To enable efficient computational evaluation without relying on human ratings, we operationalized the concept of "distinct" as the difference between the distribution $Q$ of a generative model (e.g., images generated by StableDiffusion in TWIGMA) and the distribution $P$ of real data (e.g., real images in LAION [42]); we then leverage recent advances in quantifying the difference between two continuous distributions. Our analysis incorporated a diverse range of image datasets, including AI-generated images from TWIGMA and DiffusionDB, as well as human images from LAION [42] and ArtBench [20] (see details in Table 1). ArtBench features 60,000 high-quality, genre-annotated images of artwork spanning ten different art genres from the 14th century to the 21st century; and LAION encompasses CLIP-similarity-filtered image-text pairs sourced from the Common Crawl.

To investigate the novelty of AI-generated images, we first employed two-dimensional UMAP [25] in the CLIP embedding space to visualize the differences in distributions between AI-generated and non-AI-generated images. Additionally, we used $k$-means clustering (with the efficient implementation in `FAISS` [13]) and computed Kullback-Leibler (KL) divergence among pairs of image distributions via the quantization approach outlined in Pillutla et al. [30]; our results are robust to the choices of divergence measures and specific estimators thereof.

**Measuring the variation of AI-generated images:**  Quantifying the variation of a continuous distribution is a well-studied topic across many disciplines. In this paper, we primarily drew upon existing literature that assesses diversity in natural images and generated texts: (i) pairwise distance,

computed as $\sum_{i \neq j} \|x_i - x_j\|_2 / N(N-1)$, where $x_i, x_j$ are embeddings of randomly sampled images; this metric is equivalent to cosine distance when $x_i$ and $x_j$ have unit $\ell_2$ norms; (ii) inverse of explained variance (IEV) by the largest singular value [54], $\sum_{i=1}^{d} \sigma_d / \sigma_1$, where $\sigma_i$ denotes the $i$th-largest singular value of the embedding matrix; (iii) product of marginal variances (PMV) [17], $\left( \prod_{i=1}^{d} \hat{s}_i \right)^{1/d}$, where $\hat{s}_i$ denotes the sample standard deviation of the $i$th feature; and (iv) entropy estimated using a quantization approach [35]. We adjusted the original definitions slightly for some measures to ensure that a *higher* value always indicates *greater variability* in the data.

**Characterizing the themes of AI-generated images:** To characterize the themes of AI-generated images posted on Twitter, we first applied $k$-means clustering to group images in TWIGMA. Subsequently, we employed the BLIP [19] model to caption the images in TWIGMA, which facilitated the interpretation of the obtained image clusters. To assess the quality of the theme revealed by the BLIP captions, we conducted an additional experiment with four volunteer annotators (two males, and two females; two of them have used generative text-to-image tools). The volunteers were tasked to: (1) annotate randomly-sampled images (10 from each identified cluster) with three relevant keywords; and (2) devise groupings for the images without knowledge of the clustering we had established.

## 4 Results

### 4.1 Quantified novelty and variations of AI-generated images

In this section, we present our analysis of the novelty and variations of AI-generated images. Figure 2 displays different approaches we used to compare AI-generated and non-AI-generated images. The UMAP density plot in Figure 2(a) reveals minimal overlap between sampled LAION, TWIGMA, and ArtBench images, whereas the overlap between TWIGMA and DiffusionDB is substantial. We note a distinct density peak of TWIGMA images absent in DiffusionDB, primarily representing a subset of images generated in specific styles (such as anime) and often of NSFW nature (see Figure 4 for more details). This distinction is further confirmed by estimated $k$-means clusters and KL divergence between image distributions, as shown in panels (b) and (c) of Figure 2. Specifically, cluster 1 primarily consists of LAION images, while AI-generated images are present in clusters 2–4. Additionally, the estimated KL divergence between AI-generated and non-AI-generated data pairs tends to be much larger compared to the divergence between DiffusionDB and TWIGMA.

While panels (a)–(c) in Figure 2 focused on examining whether AI-generated and non-AI-generated images are distinct, we additionally investigated whether image distances contain useful information about an image's likability. Figure 2(d) presents the analysis results for TWIGMA data (limited to the subset for which we were able to obtain likes data using Twitter API): On the left side, we present the average similarities of the five nearest neighbors in LAION and ArtBench, determined using cosine similarities, for the images in TWIGMA. On the right side, we plot the average likes for each quintile of similarities, where quintiles 1 and 5 represent the least and most similar images to LAION or ArtBench, respectively. There is some evidence suggesting that AI-generated images that are less similar to their non-AI-generated counterparts receive more likes. For instance, quintile 1 images for LAION and Artbench received, on average, 30.1 (95% CI: [28.8, 31.5]) and 31.9 (95% CI: [29.7, 34.1]) likes, respectively, compared to 23.8 (95% CI: [21.9, 25.7]) and 24.1 (95% CI: [21.5, 26.7]) likes in quintile 5. However, it's important to note that the number of likes for a tweet is influenced by various factors, such as a user's social networks [11, 49]. To account for these factors, we used the following Poisson regression that incorporates the number of followers and similarity quintiles:

$$\log(\mathbb{E}(\text{Likes}|X)) = \alpha + \beta_1 \mathbb{1}(\text{500-5,000 followers}) + \beta_2 \mathbb{1}(\text{>5,000 followers}) + \sum_{i=1}^{4} \gamma_i \mathbb{1}(\text{Quintile } i). \quad (1)$$

The regression model in (1) shows that within the same follower count category (i.e., $< 500, 500 - 5,000$, or $> 5,000$ followers), compared to TWIGMA images most similar to their ArtBench neighbors (quintile 5), images in quintile 1 receive, on average, 19% more likes (95% CI: [18%, 19%], $p<0.0001$); similar trends are observed for images in quintile 2 (12% more likes; 95% CI: [11%, 12%]; $p<0.0001$), while the differences are less pronounced for more similar quintiles (0.8% fewer likes in quintile 3 and 2% more likes in quintile 4). Comparable results are found for LAION neighbors as well, where images in quintiles 1 and 2 receive, on average, 10% (95% CI: [9%, 10%], $p<0.0001$) and 2% more (95% CI: [2%, 3%], $p<0.0001$) likes, respectively, compared to quintile 5 images posted by accounts with the same follower categories. Upon further investigation, we observe

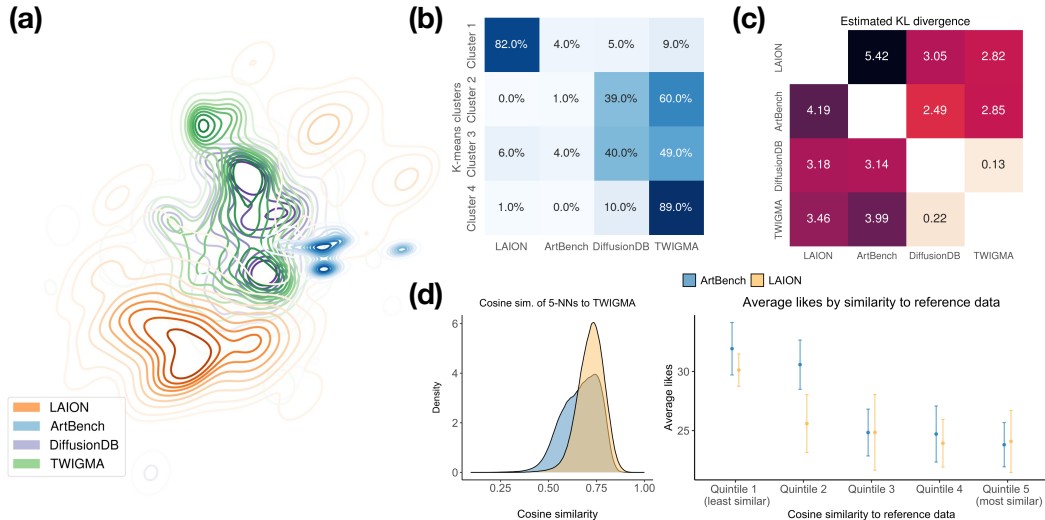

Figure 2: **Comparing AI-generated and non-AI-generated images. (a):** Kernel density plot of the 2D-UMAP embedding of randomly-sampled images from TWIGMA, LAION, DiffusionDB, and ArtBench. **(b):** K-means clustering (with $k = 4$) separates AI-generated and non-AI-generated images. **(c):** Estimated KL divergence of pairs of image distributions using the quantization approach (with 10 clusters) outlined in Section 3. Each entry in the table represents the KL divergence between the row and column distributions, e.g., the 5.42 entry is the KL divergence between LAION and ArtBench, defined as $\sum_x p_{\text{LAION}}(x) \log \left( p_{\text{LAION}}(x)/p_{\text{ArtBench}}(x) \right)$. **(d):** (Left) Average cosine similarity of the five nearest neighbors in LAION/ArtBench for TWIGMA images. (Right) Average number of likes per similarity quintile, with error bars representing 95% confidence intervals. On average, TWIGMA images least similar to LAION/ArtBench receive the most likes.

that many of the highly-liked images that stand out from the non-AI-generated images are often anime-style NSFW photos. This observation highlights the need for caution when interpreting the number of likes as an additional indicator of aesthetic and creative value for an image.

Next, we explore image variations across distributions (see Figure 3(a)–(b)). On average, LAION images exhibit the highest variability, followed closely by ArtBench, DiffusionDB, and TWIGMA, which show similar variation distributions based on pairwise Euclidean distance. Notably, ArtBench shows reduced variation when conditioned on image pairs from the same artist. Similarly, conditioning on prompts in DiffusionDB reduces around 50% of the variation in generated images, measured by the average distance between two image embeddings; that is, the average distance between images in DiffusionDB with the same prompt is roughly half that between randomly selected images with different prompts. Additional variation metrics in Figure 3(b) align with pairwise distance. The lower estimated entropy for ArtBench may be due to its smaller sample size compared to the other datasets.

Given the observation that a substantial portion of the variations in the image output space is explained by the variation in the input prompts, we further analyzed the latter in Figure 3(c)–(d). Specifically, in (c), we plotted the average pairwise distance of images with the same prompt as a function of the number of words in the prompts. Each dot represents a unique prompt, and we observe that, on average, longer prompts with more details lead to reduced variations in the output images (Kendall's $\tau$: -0.1, $p$-values: 0.01). Furthermore, to calibrate the variation of the prompt variations, we compared the pairwise distance of CLIP embeddings from different text input sources with similar word counts to the DiffusionDB prompts. These sources included real image captions [45], English Haikus [1], Twitter text (after removing hyperlinks and hashtags), CNN news summarization [43], and a book chapter [55]. We found that image captions exhibit similar variation to DiffusionDB prompts and Twitter texts, indicating higher variability. In contrast, as expected, texts with unified styles and themes, such as Haikus and book chapters, show significantly lower variation. Overall, our analysis suggests a wide range of topics covered by the DiffusionDB prompts.

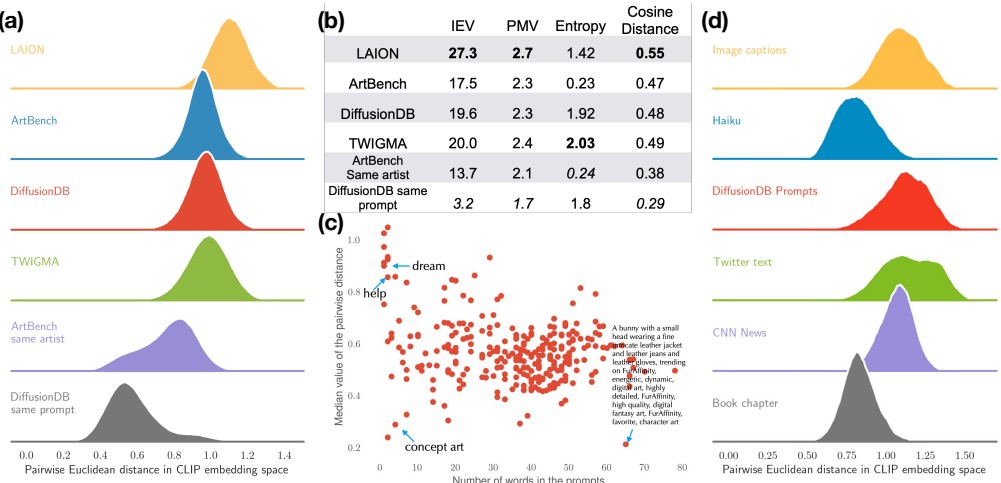

**Figure 3: Comparing variations of AI-generated and non-AI-generated images. (a):** Pairwise Euclidean distance of randomly-sampled, $\ell_2$-normalized image embeddings from LAION, ArtBench, DiffusionDB, TWIGMA, paintings by the same artists in ArtBench, and DiffusionDB images with identical prompts. **(b):** Additional variation metrics (see Section 3 for details) for the dataset mentioned in (a); larger values of these metrics correspond to a more variable dataset. Metric values for the most and least variable datasets are in **bold** and *italic*, respectively. **(c):** Average pairwise Euclidean distance of $\ell_2$-normalized embeddings in DiffusionDB with identical prompts (each dot is a unique prompt); on average, longer and more detailed prompts correspond to reduced variations in output images. **(d):** Similar analysis as in (a), but with sampled text embedding data from real image captions, English Haikus, DiffusionDB prompts, Tweets, CNN news, and a book chapter.

## 4.2 Themes of AI-generated images on Twitter

Here, we examine the themes and their longitudinal changes for images in the TWIGMA dataset. In Figure 4(a), we display the creation-date color-coded two-dimensional UMAP of TWIGMA image embeddings, suggesting a shift in the image distribution over time in our Twitter dataset. To investigate this qualitative observation further, we applied $k$-means clustering to the TWIGMA data with $k = 10$ and plotted the change of cluster membership over time. We note substantial changes in cluster memberships: clusters 1, 3, and 4 have experienced a steady decline over time, while clusters 8 and 9 showed a consistent increase. Here, the choice of $k = 10$ was motivated by the observation in Figure 3(a), which indicated that TWIGMA exhibits comparable variation to ArtBench, a dataset with 10 distinct art styles (sensitivity analysis using other values of $k$ yielded similar results).

In Figure 4(c), we visualize each cluster along with the most frequent topics derived using BLIP captions, with emojis representing the relative trends over time. Prominent themes we observed include painting, woman, and man, aligning with the known interest of users in generating detailed human portraits in various styles using text-to-image models [15, 48, 53]. Our findings also indicate a shift in preferences for image topics among Twitter users over time: there is a growing interest in sharing artistically sophisticated or distinct content, such as intricate human portraits, while interest in simpler themes such as natural scenes has declined. In addition, clusters 5 and 8 notably contain a substantial number of images with increasing popularity but also significant amounts of NSFW, pornographic, and nude content. This observation aligns with recent studies that highlight the rapid growth of online communities focused on generating such contents [14, 34, 41]. Inspired by this observation, we obtained predicted NSFW scores using a pre-trained detector [4], revealing a substantial number of likely NSFW images in both TWIGMA and LAION (see Figure 4(d)).

We observe that portrayed themes in AI-generated images extend beyond singular art genres and are swiftly evolving over time. For a human art example, during the Renaissance period spanning centuries, the focus remained primarily on real-life human portraits and scenes from contemporary life [50]. Similarly, modern art often features recurring motifs such as abstract shapes and vivid hues [36] (akin to examples in Clusters 1 and 7 illustrated in Figure 4(c), for instance). This observation on the fast-evolving themes in AI-generated images finds support in efforts that decentralize and

individualize diffusion models through LoRA updates [12, 39], resulting in numerous model variants characterized by specific themes within a span of just under a year.

We also evaluate the concordance between themes extracted from captioning models and those pinpointed by human annotator volunteers, as shown in Figure 4(c). Notably, a substantial alignment emerges between themes uncovered via unsupervised learning (middle column) and those surfaced through the annotators' keyword-based approach (right column), with overlapping themes highlighted in bold. While the initial annotator pool is small, feedback from the volunteers shed useful insights: for instance, the two NSFW clusters primarily comprise explicit content of females, and therefore are challenging to differentiate for the human annotators. Moreover, comparing human grouping with $k$-means clustering unveils that images allocated to clusters 1, 2, and 7 exhibit substantial heterogeneity, often leading to their segregation into distinct clusters when annotated by humans.

Lastly, we used cosine similarity to extract the nearest neighbors of TWIGMA images and identified pairs with a high likelihood that the neighbors from ArtBench and LAION served as potential inspirations (i.e., training data) for the text-to-image models (see Figure 4(e)). The striking similarities between some of these neighbor pairs suggest the potential of leveraging a similarity-measure-based approach to more systematically identify such pairs.

## 5 Discussion

Recent advancements in generative AI have revolutionized text-to-image generation, empowering users to create millions of captivating images. Our work explores themes and variations in AI-generated images. To facilitate this investigation, we introduce TWIGMA, an extensive dataset comprising 800,000 gen-AI images, associated tweets, and metadata collected from Twitter between January 2021 and March 2023. Our analysis characterizes distinctiveness, variation, and longitudinal shift of themes of gen-AI images shared on Twitter. The analysis in this paper is not meant to be exhaustive but rather to illustrate the types of interesting questions that TWIGMA can help to answer, opening the door to investigating various facets of human-AI art generation.

**Limitations:** It is important to note the limitations in our work, primarily due to the large-scale nature of the datasets employed. One limitation stems from the scope and quality of the datasets utilized in our analysis. Although we made efforts to select representative data sources from both AI-generated and non-AI-generated image spaces, the content within the final datasets used in our paper imposes certain constraints: ArtBench primarily consists of popular art genres in WikiArt [2], resulting in limited coverage of non-European, modern arts, as well as works from lesser-known independent artists. Consequently, this could lead to an underestimate of the similarity between human art images and AI-generated art images. Similarly, LAION data was pre-filtered based on text-image pair similarity [42], and around 10% of Twitter images were not available through the official Twitter API at the time of our study, due to the deletion or removal of tweets. Additionally, it is possible that a subset of images within TWIGMA are non-AI-generated, as users sometimes share relevant non-AI-generated content, such as real images closely resembling AI-generated outputs or screenshots from model websites or APIs. All of the aforementioned data issues could bias our findings in Section 4.1. Lastly, it is essential to acknowledge that while TWIGMA benefits from an extended time frame and Twitter's substantial user base in comparison to prior research efforts, the images in TWIGMA only constitute a subset of those publicly shared on a specific channel. When interpreting outcomes drawn from this dataset, researchers should bear in mind that the user base inherent to TWIGMA may potentially differ from their demographic of interest.

**Future work:** There are several promising avenues for future research. Firstly, the inclusion of more contemporary and modern art images, particularly illustration art and anime, in our analysis, along with the identification of non-AI-generated images in TWIGMA using state-of-the-art methods [7, 46], would strengthen the robustness of our findings. Moreover, maintaining regular updates to the TWIGMA dataset in order to consistently monitor the trajectory of themes and contents of AI-generated images would provide valuable insights for the research community. Furthermore, introducing a more human-centric perspective into dataset curation and analysis could enrich downstream insights. For instance, our current emphasis on variations in the CLIP embedding space could be complemented by exploring other dimensions of diversity and variation [24, 29]. This could entail investigating stereotypical associations between output images and societal representations within human images (e.g., whether generated NSFW images are disproportionately associated with

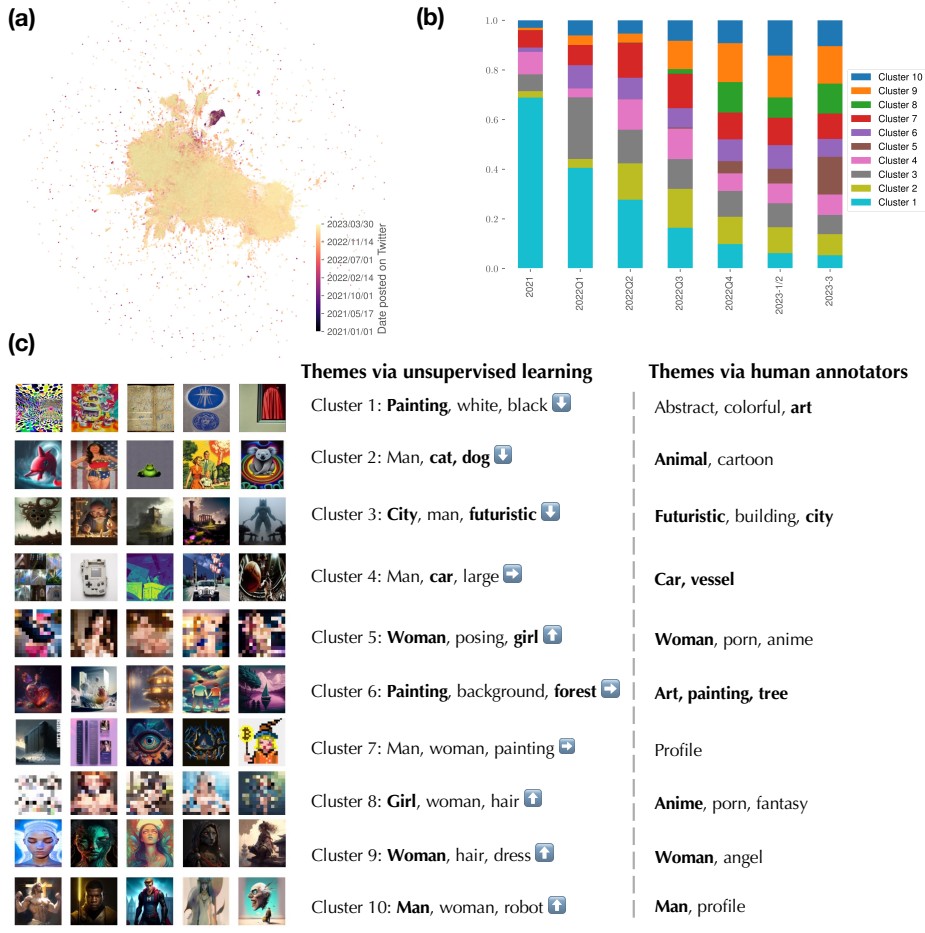

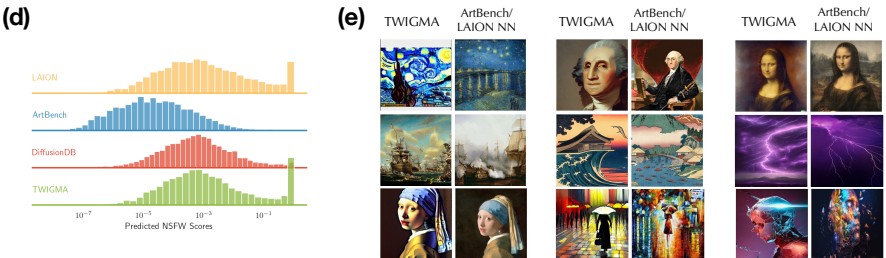

Figure 4: **Themes and longitudinal trends of images in TWIGMA. (a):** Two-dimensional UMAP embedding of TWIGMA images, color-coded based on their creation dates on Twitter. **(b):** Composition of image clusters (estimated using $k$-means clustering with $k = 10$) over time. We observe notable changes in cluster membership and underlying themes of TWIGMA images from 2021 to 2023. **(c):** *Left*: Randomly sampled images from the 10 clusters identified in (b). Clusters 5 and 8 predominantly contain NSFW photos and have been pixelated accordingly. *Middle*: Cluster annotations derived from the most frequently occurring words in the BLIP-inferred captions. Upward, downward, and rightward arrow emojis signify rising, declining, or consistent trends over time, respectively. *Right*: Cluster annotations obtained by human annotators. Themes that align between BLIP and human annotations are highlighted in bold. **(d):** Predicted NSFW scores from a pretrained CLIP-based NSFW detector. Noteworthy presence of NSFW photos is observed in both TWIGMA and LAION datasets. **(e):** TWIGMA images and their nearest neighbors in terms of image embedding cosine similarities from ArtBench or LAION. These images from ArtBench and LAION serve as potential inspirations (i.e., training data) for the text-to-image model, indicating the possibility of identifying such pairs using a similarity-measure-based approach.

one race or gender group). As another example, one potential follow-up study could leverage a large pool of human annotators to more comprehensively investigate two pivotal aspects: (i) the alignment between human comprehension and categorization of TWIGMA images and scalable machine learning outputs; and (ii) the correlation between human ratings and the number of likes for an image (included as part of TWIGMA). These studies could empower the development of future generative AI models that aim to align more closely with specific human preferences and image styles.

**Safety and ethical concerns:** One challenging aspect of text-to-image generative models is the generation of NSFW content. Our data analysis revealed a substantial subset of explicit images posted on Twitter, even among those with high likes and retweets. While some generative models like DALLE2 and StableDiffusion have built-in safety filters that block generated NSFW images, these filters can still be circumvented through prompt engineering [34, 41]. Moreover, there is a subcategory of models and online communities specifically dedicated to generating NSFW content. These observations serve as a cautionary tale for large-scale studies involving AI-generated content, as NSFW content is likely to be present and popular among certain subsets of followers due to the relatively low regulation in the current landscape. Additionally, our analysis also highlighted some AI-generated images that bear a striking resemblance to images found in human images and art datasets. These findings bear implications for potential copyright violations if training images were used without explicit consent, especially when the outputs are essentially reproductions or minor edits of memorized training examples. Finally, there is a risk of perpetuating stereotypical representations if AI-generated images tend to resemble specific groups based on demographic identities. Similar observations, discussions, and potential mitigations have been noted in other recent works [10, 24].

## 6   Conclusion

Understanding themes, contents, and user interests of AI-generated images is a critical topic that requires data beyond the currently available prompt and output image pairs datasets. We introduced TWIGMA, an extensive dataset comprising 800,000 gen-AI images, tweets, and associated metadata from Twitter (Jan 2021–March 2023). Our contribution is two-fold: Firstly, TWIGMA enables analysis of AI-generated image content and evolution across models and time on Twitter. Additionally, our analysis highlights the distinctiveness and variation of AI-generated images when compared to non-AI-generated content. We hope that our dataset and analysis will contribute to the broader discourse on the safety, novelty, and sociological or legal challenges posed by AI-generated images.

## Acknowledgments and Disclosure of Funding:

We thank Zhi Huang for sharing their code to scrap Twitter content using the public Twitter API. YC is supported by a Stanford Data Science Postdoctoral Fellowship. JZ is supported by the National Science Foundation (CCF 1763191 and CAREER 1942926), the US National Institutes of Health (P30AG059307 and U01MH098953) and grants from the Silicon Valley Foundation and the Chan-Zuckerberg Initiative.

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
