# Supplementary Materials for
# TWIGMA: A dataset of AI-Generated Images with Metadata From Twitter

**Yiqun T. Chen**
Department of Biomedical Data Science
Stanford University
Stanford, CA 94305
yiqunc@stanford.edu

**James Zou**
Departments of Biomedical Data Science;
Electrical Engineering; and Computer Science
Stanford University
Stanford, CA 94305
jamesz@stanford.edu

## A Details for dataset distribution

We have provided additional details regarding TWIGMA on our website at https://yiqunchen.github.io/TWIGMA/. The dataset used for analysis can be downloaded from https://zenodo.org/record/8031785. Additionally, readers may find our interactive introduction at https://huggingface.co/spaces/yiquntchen/TWIGMA informative. We confirm that as authors of TWIGMA, we will provide necessary maintenance, such as addressing questions and investigating potential bugs related to the dataset.

As per Twitter's official policy on data usage and sharing, we want to emphasize that we have not included any raw Twitter text or media in the dataset. Only the Twitter IDs are provided. If users wish to download and review the original Twitter posts, they should access the source page directly on Twitter and *fully comply* with the rules and regulations outlined in the *official Twitter developer policy*, available at https://developer.twitter.com/en/developer-terms/policy. The TWIGMA dataset, which consists of non-personally-identifiable, no-raw-Twitter-content, has been released under a Creative Commons Attribution 4.0 International License.

Furthermore, it is important to note that while we have taken measures to exclude potentially personally identifiable information, such as user IDs, from the dataset, a small number of tweets might still contain context about individuals' activities and locations at the time of posting. While previous research has established that the majority of public Twitter messages avoid disclosing sensitive information like phone numbers, email addresses, and residential details [1–3], we strongly advocate for the responsible utilization of TWIGMA. In particular, we urge users to adhere to the Twitter privacy policy and respect users' privacy preferences. For comprehensive details on this matter, please refer to https://twitter.com/en/privacy.

Lastly, in Section 5 of our paper, we extensively discuss the presence of a significant amount of NSFW (not-safe-for-work) content within the TWIGMA dataset. It includes content that is violent, pornographic, or contains nudity. Since our aim is to explore the content and themes of AI-generated images without filtering, we have included two fields in the final TWIGMA dataset: possibly_sensitive, which is a binary indicator of whether Twitter classifies an image as sensitive content, and nsfw_score, which represents the predicted NSFW score from a pre-trained CLIP-based NSFW detector (where a score of 1 indicates a higher likelihood of being NSFW). These fields can help identify and handle these types of images appropriately.

## B Datasheet for datasets

**For what purpose was the dataset created?** Was there a specific task in mind? Was there a specific gap that needed to be filled? Please provide a description.

The creation of large-scale image-text datasets has rapidly evolved in the past few years. As text-to-image models continue to demonstrate unprecedented capabilities in generating images based on user prompts, researchers have started curating datasets featuring images generated by these models. However, most of the existing datasets are limited in terms of style (as they often originate from a single model variation), user distribution (restricted to users of specific channels or APIs), and relatively short data collection periods (typically within a month); see our detailed discussion of prior work in Section 2 of our paper. Therefore, we propose TWIGMA, a large-scale dataset encompassing 800,000 AI-generated images from diverse models, in this paper. Spanning January 2021 to March 2023, TWIGMA covered an extended timeframe and included valuable metadata, such as inferred image subjects and number of likes. To the best of our knowledge, TWIGMA is the first AI-generated image dataset with substantial time span and rich metadata, enabling analysis of temporal trends in human-AI generated image content.

**Who created this dataset (e.g., which team, research group) and on behalf of which entity (e.g., company, institution, organization)?**

The first author of this paper (YC) curated the dataset under the supervision of the senior author (JZ); both authors are affiliated with Stanford University.

**Who funded the creation of the dataset?** If there is an associated grant, please provide the name of the grantor and the grant name and number.

YC is supported by a Stanford Data Science Postdoctoral Fellowship. JZ is supported by the National Science Foundation (CCF 1763191 and CAREER 1942926), the US National Institutes of Health (P30AG059307 and U01MH098953) and grants from the Silicon Valley Foundation and the Chan-Zuckerberg Initiative.

Composition

**What do the instances that comprise the dataset represent (e.g., documents, photos, people, countries)?** Are there multiple types of instances (e.g., movies, users, and ratings; people and interactions between them; nodes and edges)? Please provide a description.

TWIGMA contains images, texts, and associated metadata from Twitter (Jan 2021–March 2023). However, per data sharing policy from Twitter, we will only be able to include Twitter id and the derived metadata in our final dataset. See details at `https://yiqunchen.github.io/TWIGMA/`.

**How many instances are there in total (of each type, if appropriate)?**

During our analysis, we utilized a total of 805,650 unique images. It is important to acknowledge that due to the dynamic nature of Twitter, certain content may have been deleted or set to private since our analysis was conducted. Consequently, conducting a similar analysis at a later time will naturally result in a reduced number of accessible or downloadable images.

**Does the dataset contain all possible instances or is it a sample (not necessarily random) of instances from a larger set?** If the dataset is a sample, then what is the larger set? Is the sample representative of the larger set (e.g., geographic coverage)? If so, please describe how this representativeness was validated/verified. If it is not representative of the larger set, please describe why not (e.g., to cover a more diverse range of instances, because instances were withheld or unavailable).

Our objective is to curate a comprehensive dataset of AI-generated images on Twitter by collecting tweets that include at least one of the 19 hashtags listed in Figure 1 of the main text. It is important to note that there is a possibility of omitting some images that were posted without using any of these hashtags.

**What data does each instance consist of? "Raw" data (e.g., unprocessed text or images) or features?** In either case, please provide a description.

TWIGMA contains images, texts, and associated metadata from Twitter (Jan 2021–March 2023). However, per data sharing policy from Twitter, we will only be able to include Twitter id and the derived metadata in our final dataset. See details at `https://yiqunchen.github.io/TWIGMA/`.

**Is there a label or target associated with each instance?** If so, please provide a description.

N/A.

**Is any information missing from individual instances?** If so, please provide a description, explaining why this information is missing (e.g., because it was unavailable). This does not include intentionally removed information, but might include, e.g., redacted text.

Due to the dynamic nature of Twitter, content may have been deleted or set to private during the course of our analysis. Consequently, metadata such as likes is missing for those contents that become unavailable to the public.

**Are relationships between individual instances made explicit (e.g., users' movie ratings, social network links)?** If so, please describe how these relationships are made explicit.

N/A.

**Are there recommended data splits (e.g., training, development/validation, testing)?** If so, please provide a description of these splits, explaining the rationale behind them.

We do not have recommended splits, but want to mention that we do observe a change in the underlying content temporally.

**Are there any errors, sources of noise, or redundancies in the dataset?** If so, please provide a description.

It is possible that a small proportion of images within TWIGMA are non-AI-generated, as users sometimes share relevant non-AI-generated content, such as real images closely resembling AI-generated outputs or screenshots from model websites or APIs. In addition, while we deduplicated our datasets based on media id and image embedding, there still could be a small set of near duplicates of the same images in our dataset.

**Is the dataset self-contained, or does it link to or otherwise rely on external resources (e.g., websites, tweets, other datasets)?** If it links to or relies on external resources, a) are there guarantees that they will exist, and remain constant, over time; b) are there official archival versions of the complete dataset (i.e., including the external resources as they existed at the time the dataset was created); c) are there any restrictions (e.g., licenses, fees) associated with any of the external resources that might apply to a future user? Please provide descriptions of all external resources and any restrictions associated with them, as well as links or other access points, as appropriate.

Due to the official data sharing policy prescribed by Twitter, we cannot share the original content (text and images) from Twitter. Therefore, we expect that a subset of the data included in our data will not be available for retrieval, as users hide and delete their contents (as well as moderation efforts put forth by Twitter). By the time we finalize our study, less than 10% of Twitter images became unavailable through the official Twitter API due to the deletion or removal of tweets. The development of this dataset has been done in compliance with Twitter's policy on data usage and sharing. If users intend to review the original Twitter post, we recommend accessing the source page directly on Twitter and closely adhering to the official Twitter developer policy, available at `https://developer.twitter.com/en/devel oper-terms/policy`.

**Does the dataset contain data that might be considered confidential (e.g., data that is protected by legal privilege or by doctor-patient confidentiality, data that includes the content of individuals non-public communications)?** If so, please provide a description.

The development of this dataset has been done in compliance with Twitter's policy on data usage and sharing. The use of this dataset is solely at your own risk and should be in accordance with applicable laws, regulations, and ethical considerations. If you intend to review the original Twitter post, we recommend accessing the source page directly on Twitter and closely adhering to the official Twitter developer policy, available at `https://developer.twitter.com/en/devel oper-terms/policy`.

**Does the dataset contain data that, if viewed directly, might be offensive, insulting, threatening, or might otherwise cause anxiety?** If so, please describe why.

It is important to note that a substantial amount of images in this dataset have been classified as NSFW (not-safe-for-work) by both Twitter and a CLIP-based NSFW model. This includes content that is violent, pornographic, or contains nudity. We have chosen not to exclude these images from the dataset in order to understand the content and themes without filtering. However, we have included two fields in the final TWIGMA dataset

`possibly_sensitive` and `nsfw_score` which can be used to filter out these images.

**Does the dataset relate to people?** If not, you may skip the remaining questions in this section.

`N/A` since only Twitter ids are provided.

**Does the dataset identify any subpopulations (e.g., by age, gender)?** If so, please describe how these subpopulations are identified and provide a description of their respective distributions within the dataset.

`N/A`.

**Is it possible to identify individuals (i.e., one or more natural persons), either directly or indirectly (i.e., in combination with other data) from the dataset?** If so, please describe how.

`N/A`.

**Does the dataset contain data that might be considered sensitive in any way (e.g., data that reveals racial or ethnic origins, sexual orientations, religious beliefs, political opinions or union memberships, or locations; financial or health data; biometric or genetic data; forms of government identification, such as social security numbers; criminal history)?** If so, please provide a description.

`N/A`.

---

### Collection Process

**How was the data associated with each instance acquired?** Was the data directly observable (e.g., raw text, movie ratings), reported by subjects (e.g., survey responses), or indirectly inferred/derived from other data (e.g., part-of-speech tags, model-based guesses for age or language)? If data was reported by subjects or indirectly inferred/derived from other data, was the data validated/verified? If so, please describe how.

TWIGMA contains the following fields:

- id: This is the Twitter id uniquely identifying each tweet used in this dataset and our analysis;
- image_name: This is the media id used to uniquely identify each photo. Leveraging this field is necessary since a tweet can contain multiple images;

- created_at: This is the time of creation corresponding to the Twitter id;
- like_count: This is the number of likes collected from official Twitter API (snapshot: the week of May 29th). Note that some likes are not available because the corresponding tweets have been deleted since we first downloaded the photos;
- quote_count: Same as like_count, but for quotes;
- reply_count: Same as like_count, but for replies;
- all_captions: This is the BLIP-generated (Li et al. 2022) captions for the corresponding image;
- label_10_cluster: This is the assigned k-means cluster (k=10 so this number varies from 1 to 10);
- possibly_sensitive: Binary variable indicating whether the media content has been marked as sensitive/NSFW by Twitter;
- nsfw_score: The predicted NSFW from a pre-trained CLIP-based NSFW detector (ranges from 0 to 1; closer to 1 means more likely to be NSFW);
- UMAP_dim_1: The first dimension for a two-dimensional UMAP projection of the CLIP-ViT-L-14 embeddings of the images in TWIGMA.
- UMAP_dim_2: The second dimension for a two-dimensional UMAP projection of the CLIP-ViT-L-14 embeddings of the images in TWIGMA.

Out of these fields, `id`, `image_name`, `created_at`, `like_count`, `quote_count` and `reply_count` are directly observable variables from Twitter. On the other hand, `all_captions` is derived from a BLIP (deep learning) model where we validated the outcome by manually inspecting random pairs of images and captions; `possibly_sensitive` and `nsfw_score` are two predicted scores indicating how sensitive/NSFW an image might be. Finally, `label_10_cluster` is derived from k-means clustering and `UMAP_dim_1/2` are derived from a UMAP projection.

**What mechanisms or procedures were used to collect the data (e.g., hardware apparatus or sensor, manual human curation, software program, software API)?** How were these mechanisms or procedures validated?

We used Twitter API and employed an sequential approach to curate the TWIGMA data (see details in Section 3 of the main text). The image embeddings were performed on a single GPU with 32GB RAM and the UMAP embeddings were performed on multi-core CPUs with 256GB RAM; both are conducted over computing clusters.

**If the dataset is a sample from a larger set, what was the sampling strategy (e.g., deterministic, probabilistic with specific sampling probabilities)?**

We performend simple random sampling whenever the analysis of the full data would be too time-consuming and does not add substantial value: UMAP and density visualization, as well as samples of non-AI-generated human images.

**Who was involved in the data collection process (e.g., students, crowdworkers, contractors) and how were they compensated (e.g., how much were crowdworkers paid)?**

`N/A`, only the first author was invovled in the data collection process.

**Over what timeframe was the data collected? Does this timeframe match the creation timeframe of the data associated with the instances (e.g., recent crawl of old news articles)?** If not, please describe the timeframe in which the data associated with the instances was created.

Data present in TWIGMA ranged from Jan. 2021 to Mar. 2023. The curation process of this project took place from Jan. to May 2023.

**Were any ethical review processes conducted (e.g., by an institutional review board)?** If so, please provide a description of these review processes, including the outcomes, as well as a link or other access point to any supporting documentation.

`N/A`.

**Does the dataset relate to people?** If not, you may skip the remaining questions in this section.

`N/A`; only Twitter ids are provided.

**Did you collect the data from the individuals in question directly, or obtain it via third parties or other sources (e.g., websites)?**

`N/A`.

**Were the individuals in question notified about the data collection?** If so, please describe (or show with screenshots or other information) how notice was provided, and provide a link or other access point to, or otherwise reproduce, the exact language of the notification itself.

`N/A`.

**Did the individuals in question consent to the collection and use of their data?** If so, please describe (or show with screenshots or other information) how consent was requested and provided, and provide a link or other access point to, or otherwise reproduce, the exact language to which the individuals consented.

`N/A`.

**If consent was obtained, were the consenting individuals provided with a mechanism to revoke their consent in the future or for certain uses?** If so, please provide a description, as well as a link or other access point to the mechanism (if appropriate).

`N/A`.

**Has an analysis of the potential impact of the dataset and its use on data subjects (e.g., a data protection impact analysis) been conducted?** If so, please provide a description of this analysis, including the outcomes, as well as a link or other access point to any supporting documentation.

`N/A`.

---

**Preprocessing/cleaning/labeling**

---

**Was any preprocessing/cleaning/labeling of the data done (e.g., discretization or bucketing, tokenization, part-of-speech tagging, SIFT feature extraction, removal of instances, processing of missing values)?** If so, please provide a description. If not, you may skip the remainder of the questions in this section.

We provide simple deduplication using twitter media id as well as CLIP image embeddings. We did not impute any missing values but simply reported the results after dropping the missing values in our analysis. Images collected in our study are transformed into CLIP image embeddings and many subsequent analyses (including

the metadata such as clustering labels and UMAP coordinates) are based on these embeddings.

**Was the "raw" data saved in addition to the preprocessed/cleaned/labeled data (e.g., to support unanticipated future uses)?** If so, please provide a link or other access point to the "raw" data.

The raw data containing images and texts are not shared due to Twitter policy. We did not publicy share duplicated tweets and tweets without an image since these are not relevant to our main research questions of interest.

**Is the software used to preprocess/clean/label the instances available?** If so, please provide a link or other access point.

They are but we could not provide public access to the raw instances due to data sharing policy of Twitter. However, users can easily retrieve the raw instances using the provided Twitter id.

---

**Uses**

---

**Has the dataset been used for any tasks already?** If so, please provide a description.

We performed a comparative analysis of TWIGMA with natural images and human artwork, and found that gen-AI images possess distinctive characteristics and exhibit, on average, lower variability when compared to their non-gen-AI counterparts. We also revealed a longitudinal shift in the themes of images in TWIGMA, with users increasingly sharing artistically sophisticated content such as intricate human portraits, whereas their interest in simple subjects such as natural scenes and animals has decreased. These analyses are detailed in Section 4 of our paper.

**Is there a repository that links to any or all papers or systems that use the dataset?** If so, please provide a link or other access point.

We will release our preprint on arXiv and code on GitHub; links will be updated on the project website at https://yiqunchen.github.io/TWIGMA/.

**What (other) tasks could the dataset be used for?**

What we presented in our paper is just an initial exploration of the dataset. Users can further explore the relationship between depicted subject

and the number of likes, the underlying real images that inspired the AI-generated images, and so on so forth.

**Is there anything about the composition of the dataset or the way it was collected and preprocessed/cleaned/labeled that might impact future uses?** For example, is there anything that a future user might need to know to avoid uses that could result in unfair treatment of individuals or groups (e.g., stereotyping, quality of service issues) or other undesirable harms (e.g., financial harms, legal risks) If so, please provide a description. Is there anything a future user could do to mitigate these undesirable harms?

It is possible that the Twitter id and content in this dataset can be used to identify corresponding Twitter accounts that are active in this generative AI space, but we do not see immediate harm as long as these Twitter accounts do not reveal personally identifiable information.

**Are there tasks for which the dataset should not be used?** If so, please provide a description.

We urge that researchers exercise caution and discretion when using this dataset. The development of this dataset has been done in compliance with Twitter's policy on data usage and sharing. The use of this dataset should be in accordance with applicable laws, regulations, ethical considerations, and especially official Twitter developer policy at https://developer.twitter.com/en/developer-terms/policy.

---

**Distribution**

---

**Will the dataset be distributed to third parties outside of the entity (e.g., company, institution, organization) on behalf of which the dataset was created?** If so, please provide a description.

We distribute the metadata we collected at https://zenodo.org/record/8031785.

**How will the dataset will be distributed (e.g., tarball on website, API, GitHub)** Does the dataset have a digital object identifier (DOI)?

We release the dataset at https://zenodo.org/record/8031785; a detailed introduction to our project and dataset can be accessed at

```
https://yiqunchen.github.io/TWIGMA/.
```

**When will the dataset be distributed?**

The dataset has been made available since June 12, 2023.

**Will the dataset be distributed under a copyright or other intellectual property (IP) license, and/or under applicable terms of use (ToU)?** If so, please describe this license and/or ToU, and provide a link or other access point to, or otherwise reproduce, any relevant licensing terms or ToU, as well as any fees associated with these restrictions.

We currently released the metadata under a Creative Commons CC BY 4.0 License; if you plan to get the official Twitter content, please cosult the official Twitter policy at `https://developer.twitter.com/en/devel oper-terms/policy`.

**Have any third parties imposed IP-based or other restrictions on the data associated with the instances?** If so, please describe these restrictions, and provide a link or other access point to, or otherwise reproduce, any relevant licensing terms, as well as any fees associated with these restrictions.

Please exercise caution and discretion when using this dataset. The development of this dataset has been done in compliance with Twitter's policy on data usage and sharing. The use of this dataset is solely at your own risk and should be in accordance with applicable laws, regulations, and ethical considerations. If you intend to review the original Twitter post, we recommend accessing the source page directly on Twitter and closely adhering to the official Twitter developer policy, available at `https://developer.twitter.com/en/dev eloper-terms/policy`.

**Do any export controls or other regulatory restrictions apply to the dataset or to individual instances?** If so, please describe these restrictions, and provide a link or other access point to, or otherwise reproduce, any supporting documentation. Please exercise caution and discretion when using this dataset. The development of this dataset has been done in compliance with Twitter's policy on data usage and sharing. The use of this dataset is solely at your own risk and should be in accordance with applicable laws, regulations, and ethical considerations. If you intend to review the original Twitter post, we recommend accessing the source

page directly on Twitter and closely adhering to the official Twitter developer policy, available at `https://developer.twitter.com/en/dev eloper-terms/policy`.

| Maintenance |
|:---:|

**Who will be supporting/hosting/maintaining the dataset?**

The first author, YC, will be hosting and maintaining the dataset.

**How can the owner/curator/manager of the dataset be contacted (e.g., email address)?**

```
yiqun.t.chen@gmail.com
```

**Is there an erratum?** If so, please provide a link or other access point.

```
N/A.
```

**Will the dataset be updated (e.g., to correct labeling errors, add new instances, delete instances)?** If so, please describe how often, by whom, and how updates will be communicated to users (e.g., mailing list, GitHub)?

We do not have plans to regularly add new instances; but we will review requests and gather feedback from users on a quarterly basis.

**If the dataset relates to people, are there applicable limits on the retention of the data associated with the instances (e.g., were individuals in question told that their data would be retained for a fixed period of time and then deleted)?** If so, please describe these limits and explain how they will be enforced.

```
N/A.
```

**Will older versions of the dataset continue to be supported/hosted/maintained?** If so, please describe how. If not, please describe how its obsolescence will be communicated to users.

`N/A` since no new versions are actually being planned.

**If others want to extend/augment/build on/contribute to the dataset, is there a mechanism for them to do so?** If so, please provide a description. Will these contributions be validated/verified? If so, please

describe how. If not, why not? Is there a process for communicating/distributing these contributions to other users? If so, please provide a description.

The users can leave comments as Github issues at `https://github.com/yiqunchen/TWIGMA`.