# OpenReview forum: "TWIGMA: A dataset of AI-Generated Images with Metadata From Twitter"
_NeurIPS.cc/2023/Track/Datasets_and_Benchmarks — NeurIPS 2023 Datasets and Benchmarks Poster_

### Official Review · Reviewer_s6di · 2023-07-19
**Maybe the first well-constructed AI-Generated Images Dataset**

**Rating:** 9
**Confidence:** 3
**Correctness:** The claims seem supported by previous…
**Clarity:** The paper was well-written and concise.

**Strengths:**

- Well-written paper.
- Clear and concise structure.
- A dataset that might be the first of its kind (including gen-AI images).
- Valuable contribution in many domains dealing with gen-AI, its effects, and overall impact.

**Additional Feedback:**

Really good work! I would really love to see this getting published and utilized by the scientific community.

**Documentation:**

I found the documentation complete.

**Ethics:**

The authors provided a dedicated section explaining ethical considerations, also they have done that throughout the paper at specific points.

**Limitations:**

The limitations were adequate enough, at least in my view.

**Opportunities For Improvement:**

An improvement would be maybe if the authors could manage to provide a more human-centric aspect as metadata on those images, by including humans-in-the-loop to attribute a set of tags on it and a brief description for the gen-AI images.

**Relation To Prior Work:**

Differences with prior work were adequately discussed.

**Summary And Contributions:**

In this work, the authors provide a dataset consisting of AI-generated images and metadata gathered from Twitter's public API. Among others, they found that gen-AI images possess distinctive characteristics and exhibit lower variability when compared to their non-gen-AI counterparts. Additionally, they found that the similarity between gen-AI images and natural images is inversely correlated with the number of likes, which is an indicator of identifying human images served as an inspiration to generate those. Finally, they identified that users increasingly sharing artistically sophisticated content, whereas postings in simple subjects such as natural scenes and animals have decreased.

---

> ### Author Response · Authors · 2023-08-15
>
> Dear Reviewer s6di, thank you for your enthusiastic engagement with our research!
>
> In response to your comments, we have augmented our study to incorporate a human-centric perspective into the dataset analysis. In particular, we conducted a small-scale pilot study involving four volunteers to assess the inferred themes of the TWIGMA dataset based on the clustering BLIP captioning approach.
>
> In this study, we randomly selected 10 images from each of the 10 distinct clusters, resulting in a total of 100 images. The volunteers were tasked with two objectives: 1. Annotate each image with three relevant keywords, and 2. Devise groupings for the images without any knowledge of the k-means clustering employed in our analysis.
>
> Our findings illustrate a notable congruence between keywords and clusters as assigned by human annotators and the automatic unsupervised learning approach. This alignment is now visually represented in the revised Figure 3(c). Although our initial sample size for the human-in-the-loop experiment is modest, the insights garnered have been invaluable. Notably, the annotators highlight that clusters identified by the automatic supervised learning method can exhibit varying degrees of heterogeneity. For instance, the two NSFW clusters primarily containing explicit content of females pose a challenge for human annotators to differentiate.
>
> We have duly updated these changes into methods (refer to page x, updated section y) and results (refer to page x, updated section y) sections pertaining to this initial human-in-the-loop study in our revised submission. Additionally, we have outlined plans for a more extensive and systematic exploration of the correlation between human preferences, Twitter likes, and cluster assignments in future work (refer to page x, updated section y). We have reproduced these edits to our paper below for your convenience (please also see the revised Figure 3(c)):
>
> > **Characterizing the themes of AI-generated images:** To characterize the themes of AI-generated images posted on Twitter, we first applied $k$-means clustering to group images in TWIGMA. Subsequently, we employed the BLIP (Li et al. 2022) model to caption the images in TWIGMA, which facilitated the interpretation of the obtained image clusters. To assess the quality of the theme revealed by the BLIP captions, we conducted an additional experiment with four volunteer annotators (two males, two females; two of them have used generative text-to-image tools). The volunteers were tasked to: (1) annotate randomly-sampled images (10 from each identified cluster) with three relevant keywords; and (2) devise groupings for the images without knowledge of the clustering we had established. [*Section 3 in the revised manuscript*]
>
> > We also evaluate the concordance between themes extracted from captioning models and those pinpointed by human annotator volunteers, as shown in Figure 3(c). Notably, a substantial alignment emerges between themes uncovered via unsupervised learning (middle column) and those surfaced through the annotators' keyword-based approach (right column), with overlapping themes highlighted in bold. While the initial annotator pool is small, feedback from the volunteers shed useful insights: for instance, the two NSFW clusters primarily comprise explicit content of females, and therefore are challenging to differentiate for the human annotators. Moreover, comparing human grouping with $k$-means clustering unveils that images allocated to clusters 1, 2, and 7  exhibit substantial heterogeneity, often leading to their segregation into distinct clusters when annotated by humans. [*Section 4.2 in the revised manuscript*]
>
> > Furthermore, introducing a more human-centric perspective into dataset curation and analysis could enrich downstream insights. For instance, our current emphasis on variations in the CLIP embedding space could be complemented by exploring other dimensions of diversity and variation (Luccioni et al. 2023, Orgad et al. 2023). This could entail investigating stereotypical associations between output images and societal representations within human images (e.g., whether generated NSFW images are disproportionately associated to one race or gender group). As another example, one potential follow-up study could leverage a large pool of human annotators to more comprehensively investigate two pivotal aspects: (i) the alignment between human comprehension and categorization of TWIGMA images and scalable machine learning ouputs; and (ii) the correlation between human ratings and the number of likes for an image (included as part of TWIGMA). These studies could empower the development of future generative AI models that aim to align more closely with specific human preferences and image styles. [*Future work paragraph of Section 5*]
>
> We genuinely appreciate your constructive feedback and your ongoing interest in our research!

---

> > ### Comment · Reviewer_s6di · 2023-08-29
> >
> > I would like to thank the authors for the effort on improving their current work by addressing the comments I have provided. Based also on the rest reviewers comments/concerns, I would like to keep my score as is. I’m looking forward to see upcoming works on this in the future!

---

### Official Review · Reviewer_vkpY · 2023-07-20
**A large-scale dataset of AI-generated images with straightforward analyses**

**Rating:** 6
**Confidence:** 4
**Clarity:** The paper is well-written and easy to…

**Strengths:**

- The clarity of writing is a strength
- A large-scale dataset of 800k Twitter IDs is released alongside metadata, allowing data users to download the associated Tweets and images if they wish to conduct their own analyses. This respects Twitter's data use and sharing policies
- A temporal study of the content of AI-generated images on Twitter is conducted, which is different to previous studies introducing datasets that focus on detecting fake artworks or evaluating image quality. This can be useful for studying how people's use of AI-based technologies change over time.

**Additional Feedback:**

See opportunities for improvement.

**Correctness:**

There is no real verification that the retrieved data corresponds to AI-generated data.

**Documentation:**

Everything related to documentation is adequately addressed:

- The data collection method is clearly stated in the main text

- The paper's supplementary materials contain dataset documentation, utilizing the datasheets for datasets template. The documentation clearly warns users of Twitter's policy on data sharing and use. In addition, users are warned that the Twitter IDs may lead to NSFW content. I believe the documentation is sufficient to promote responsible use.

- Intended uses correspond to the same use cases presented in the main paper, i.e., studying the difference between AI-generated and non-AI generated images, as well as analyses of thematic changes in AI-generated data over time.

- A URL is provided to access the dataset on Zenodo

- The paper's authors take responsibility for maintaining the dataset

**Ethics:**

No.

**Limitations:**

The paper should acknowledge that Twitter users are not necessarily representative and that TWIGMA inherits biases from the platform and its users. The paper states that previous datasets are limited in terms of user distribution, however, the paper does not evidence that it is not also limited in the same way.

**Opportunities For Improvement:**

- The paper is based on the premise that we need to understand the themes, content, and diversity of AI images since some AI-generated images may, e.g., represent NSFW content. However, it is unclear to me why this dataset is needed or the significance of the analyses. While the paper is clearly written, I am unable to derive any meaningful takeaways
- RQ1 and RQ2 appear to be the same question, i.e., what are the difference between AI and non-AI generated images
- The paper states that past research [49,50] investigate topics/contents of logged prompts, which do not accurately represent the output images due to prompt engineering. However, the reverse is true of this paper, i.e., the paper does not have access to the prompts used to generate the images, therefore the automatically derived captions may not accurately represent the images.
- The paper investigates how the themes of AI images changes over time, but what is missing is a comparative analysis of how the themes of non-AI generated images changes over time.
- The paper does not verify that the images obtained from Twitter are in fact AI-generated beyond a cursory look random samples of images obtained, i.e., there are no assurances provided to dataset users that the data obtained is AI-generated. One worry is that the resulting dataset is extremely noisy
- The analyses are simplistic, comparing distributions, computing similarities in embedding spaces, and analysing the results of unsupervised clustering methods. One concern here is that there is no "human-in-the-loop" during any of the data collection or analyses.

**Relation To Prior Work:**

Yes, the paper is clear in how it differs from other similar datasets.

**Summary And Contributions:**

The paper introduces a new dataset of 800k AI-generated images obtained from Twitter, coined TWIGMA. The purpose of the dataset is to aid understanding of how AI-generated images are being used, as well as their content. To that end, the paper performs a comparative analysis of AI-generated images and non-AI-generated images, highlighting that generated images are less "novel" than real images.

---

> ### Author Response · Authors · 2023-08-15
> **Addressing Twitter user bias and "human-in-the-loop" component**
>
> We extend our sincere gratitude to Reviewer vkpY for their meticulous suggestions and insightful assessment of our paper. We highly value your valuable feedback, and we have diligently incorporated your comments and recommendations into our revised manuscript below. **Due to character limits, we will address your detailed comments in two separate messages**
>
> 1. we have taken into deep consideration the Reviewer's astute observation regarding the platform-specific nature of our proposed dataset and its alignment with a distinct user base. To address this concern, we have expanded our explanation to ensure that future utilization of the dataset remains mindful of its inherent limitations. Specifically, we emphasize in the revised manuscript:
> > Lastly, it is essential to acknowledge that while TWIGMA benefits from an extended time frame and Twitter's substantial user base in comparison to prior research efforts, the images in TWIGMA only constitute a subset of those publicly shared on a specific channel. When interpreting outcomes drawn from this dataset, researchers should bear in mind that the user base inherent to TWIGMA may potentially differ from their demographic of interest. [Limitation paragraph of Section 5 in the revision]
>
> 2. We recognize that our prior description lacked precision regarding the potential impact of curating and analyzing datasets like ours. In response, we have further clarified the broader implications of our efforts. Our research not only holds relevance from a machine learning standpoint, where we foresee the TWIGMA dataset and its future iterations catalyzing refinement variants based on the highest-liked photos and styles, but also provides insights into human interaction with generative models in a broader scientific context. As generative content continues to proliferate, our study strives to unveil insights into the output generated by data-centric AI tools, such as diffusion models. In the introduction, we have now added that
> > In addition to offering opportunities for fine-tuned adaptations to generate images with high popularity on social platforms like Twitter, our dataset and accompanying analyses contribute substantial insights into the ways in which humans engage with generative models, a topic of much interest to a broader scientific framework: as the prevalence of generative content continues to grow, our study could serve as pilot for future endeavors aimed at comprehending and enhancing the sociological [41] and human-machine interaction [10, 30] dimensions inherent to these data-driven gen-AI models.
>
> 3. Thank you for highlighting the need for clarity in the relationship between RQ1 and RQ2. We have now explicitly outlined the distinctions between the two research questions. While RQ1 delves into differences in distributional mass, RQ2 scrutinizes a specific dimension:
> > This prompts our first research question (RQ) to investigate the novelty of AI-generated images: How distinct are the distributions of AI-generated images from those of non-AI images?
> > …, and our second RQ builds upon these findings and zooms in a particular dimension of RQ1, the variation in the image space:
>
> 4. We extend our gratitude for drawing attention to an inaccurately framed statement regarding prompt versus inferred content analysis. We have rectified this statement in the revised version:
> > Moreover, we leverage unsupervised learning techniques and inferred image captions to understand themes of AI-generated images. This complements earlier research, which primarily concentrated on logged prompts' subjects and content.
>
> 5. Regarding the evolution of non-AI-generated images, we observe that themes in AI-generated images evolve much more rapidly and have updated the comment content in the revised manuscript:
> > We observe that portrayed themes in AI-generated images extend beyond singular art genres and are swiftly evolving over time. For a human art example, during the Renaissance period spanning centuries, the focus remained primarily on real-life human portraits and scenes from contemporary life [51]. Similarly, modern art often features recurring motifs such as abstract shapes and vivid hues [37] (akin to examples in Clusters 1 and 7 illustrated in Figure 4(c), for instance). This observation on the fast-evolving themes in AI-generated images finds support in efforts that decentralize and individualize diffusion models through LoRA updates [13, 39], resulting in numerous model variants characterized by specific themes within a span of just under a year. [*Revised results section*]

---

> > ### Author Response · Authors · 2023-08-15
> >
> > 6. Addressing concerns about the verification of AI-generated images from Twitter: While we acknowledge the inherent noise within our dataset due to its volume, we have taken several steps to ensure data quality, including random quality checks. Additionally, we emphasize that the distribution of TWIGMA images is overlapping with the density of DiffusionDB (see Figure 2 in the revision), indicating a strong presence of AI-generated images. We also highlighted the limitations due to the lack of comprehensive check in our manuscript:
> > > Additionally, it is possible that a subset of images within TWIGMA are non-AI-generated, as users sometimes share relevant non-AI-generated content, such as real images closely resembling AI-generated outputs or screenshots from model websites or APIs. All of the aforementioned data issues could bias our findings in Section 4.1.
> >
> > 7. Regarding the simplicity of our analyses, we recognize your concern and have acknowledged the "human-in-the-loop" perspective. While we emphasize quantitative techniques due to scalability and data volume, we have introduced a pilot human annotation study in the revised manuscript. This study engages human participants to evaluate inferred themes based on our clustering BLIP captioning approach. We have reproduced these edits to our paper below for your convenience (please also see the revised Figure 3(c)):
> >
> > > **Characterizing the themes of AI-generated images:** To characterize the themes of AI-generated images posted on Twitter, we first applied $k$-means clustering to group images in TWIGMA. Subsequently, we employed the BLIP (Li et al. 2022) model to caption the images in TWIGMA, which facilitated the interpretation of the obtained image clusters. To assess the quality of the theme revealed by the BLIP captions, we conducted an additional experiment with four volunteer annotators (two males, two females; two of them have used generative text-to-image tools). The volunteers were tasked to: (1) annotate randomly-sampled images (10 from each identified cluster) with three relevant keywords; and (2) devise groupings for the images without knowledge of the clustering we had established. [Section 3 in the revised manuscript]
> >
> > > We also evaluate the concordance between themes extracted from captioning models and those pinpointed by human annotator volunteers, as shown in Figure 3(c). Notably, a substantial alignment emerges between themes uncovered via unsupervised learning (middle column) and those surfaced through the annotators' keyword-based approach (right column), with overlapping themes highlighted in bold. While the initial annotator pool is small, feedback from the volunteers shed useful insights: for instance, the two NSFW clusters primarily comprise explicit content of females, and therefore are challenging to differentiate for the human annotators. Moreover, comparing human grouping with $k$-means clustering unveils that images allocated to clusters 1, 2, and 7  exhibit substantial heterogeneity, often leading to their segregation into distinct clusters when annotated by humans. [Section 4.2 in the revised manuscript]
> >
> > > Furthermore, introducing a more human-centric perspective into dataset curation and analysis could enrich downstream insights. For instance, our current emphasis on variations in the CLIP embedding space could be complemented by exploring other dimensions of diversity and variation (Luccioni et al. 2023, Orgad et al. 2023). This could entail investigating stereotypical associations between output images and societal representations within human images (e.g., whether generated NSFW images are disproportionately associated to one race or gender group). As another example, one potential follow-up study could leverage a large pool of human annotators to more comprehensively investigate two pivotal aspects: (i) the alignment between human comprehension and categorization of TWIGMA images and scalable machine learning ouputs; and (ii) the correlation between human ratings and the number of likes for an image (included as part of TWIGMA). These studies could empower the development of future generative AI models that aim to align more closely with specific human preferences and image styles. [Future work paragraph of Section 5]
> >
> > We genuinely appreciate your constructive feedback and your ongoing interest in our research!

---

> ### Comment · Reviewer_vkpY · 2023-08-26
> **Review response**
>
> Thank you to the authors for providing a detailed response to my review. In particular, I appreciate the effort taken to clarify statements within the revised paper.
>
> I have increased my score from 4 to 6.
>
> ---
> Note, however, that having read reviews written by reviewer VLov and reviewer Araw, I have concerns about the utility of this dataset beyond the current study. First, the future work directions do not seem as if they would garner a lot of further uses of TWIGMA. Moreover, the suggested future work directions seem like they would have been a strong contributions to the current work. Lastly, since TWIGMA is a static dataset, it's relevance will diminish rather quickly. Nonetheless, the point about the dataset being static can be applied to the majority of ML datasets, therefore I do not think the authors should be penalized for this. Thus, my main reservation is regarding the use of the dataset beyond this study.

---

### Official Review · Reviewer_8zrT · 2023-07-21
**Large-scale AI-generated twitter dataset**

**Rating:** 6
**Confidence:** 3
**Correctness:** The submission is a dataset, which is…
**Clarity:** This paper is well written and easy t…

**Strengths:**

The proposed TWIGMA dataset represents a groundbreaking advancement in the domain of AI-generated text-image datasets, which help researchers to explore and analyze temporal trends in human-AI generated image content, a capability previously unattainable in the field.

**Additional Feedback:**

Did the author desensitized the data, such as the names of people, place that may appear?

**Documentation:**

The submission provides sufficient detail on data collection and organization, availability, maintenance, documentation and intended uses

**Limitations:**

The author has adequately addressed the limitations and potential negative societal impact of their work in section 5.

**Opportunities For Improvement:**

The dataset is collected from Twitter, which may introduce biases from user portraits and content preferences.

**Relation To Prior Work:**

This is the first work of large-scale AI-generated text-image dataset.

**Summary And Contributions:**

This paper introduces the TWIGMA dataset, a vast compilation of over 800,000 AI-generated images sourced from Twitter. Employing a rigorous analytical approach, the authors examine the distinctiveness, variation, and thematic attributes inherent in these AI-generated images compared with non-AI-generated counterparts. The investigation reveals that AI-generated images exhibit distinct characteristics and lower variability.

---

> ### Author Response · Authors · 2023-08-15
>
> Thank you Reviewer 8zrT on your perceptive insights and constructive feedback on our work! We have meticulously addressed the comments and suggestions presented in your review within our revised manuscript.
>
> 1. Specifically, we have taken into account the observation that our proposed dataset is platform-specific and tailored to a distinct user base. We have elaborated on this point to ensure that any subsequent use of the dataset remains cognizant of its limitations:
> > Lastly, it is essential to acknowledge that while TWIGMA benefits from an extended time frame and Twitter's substantial user base in comparison to prior research efforts, the images in TWIGMA only constitute a subset of those publicly shared on a specific channel. When interpreting outcomes drawn from this dataset, researchers should bear in mind that the user base inherent to TWIGMA may potentially differ from their demographic of interest. [Limitation paragraph of Section 5 in the revision]
>
> 2. Moreover, we have expanded upon the topic of data sensitization in our supplementary materials, referencing prior studies that affirm the general absence of sensitive information like phone numbers, email addresses, and residential details in public Twitter messages. Simultaneously, we emphasize the importance of responsible dataset utilization in the revised supplementary materials:
> > Furthermore, it is important to note that while we have taken measures to exclude potentially personally identifiable information, such as user IDs, from the dataset, a small number of tweets might still contain context about individuals' activities and locations at the time of posting. While previous research has established that the majority of public Twitter messages avoid disclosing sensitive information like phone numbers, email addresses, and residential details (Sohail2021 et al. 2021,Humphreys et al. 2010;2014), we strongly advocate for the responsible utilization of TWIGMA. In particular, we urge users to adhere to the Twitter privacy policy and respect users' privacy preferences. For comprehensive details on this matter, please refer to https://twitter.com/en/privacy.
>
> Thank you again for your review and consideration of our work!

---

> > ### Comment · Reviewer_8zrT · 2023-08-26
> >
> > Thank you for your response, I would increase the score from 5 to 6.

---

> > > ### Author Response · Authors · 2023-08-27
> > >
> > > Thank you for your reviews and the updated rating! We are certainly excited about the possibility of extending the data source of our work from Twitter (now X) to more platforms and channels in the future.

---

### Official Review · Reviewer_Araw · 2023-07-21
**A good paper that needs improvements on several aspects**

**Rating:** 7
**Confidence:** 3

**Strengths:**

The longitudinal study is an important feature of this dataset, as well as the presence of metadata from the tweets. In addition, the proposed research questions are very interesting (RQ2 could be better phrased).

**Additional Feedback:**

The authors used the wrong template for this submission. In the revision, please fix the template to the one with the line numbers on the right-hand side. This is the one for camera-ready papers and it does not allow to refer to specific lines easily.

I am happy to participate to the open discussion for this paper and I will surely reconsider my (currently) low mark if needed. However, I will have no internet access from August 2nd to August 15th. I will resume back to work on August 16th and will be able to answer only after that date.

**Clarity:**

The paper needs some revision in the writing. In particular, the "consecutio temporum" is slightly confused and confusing. Some parts have present tenses and others have past tenses, and the logic behind it is not clear.

**Correctness:**

The dataset is constructed in a solid way. The analysis on the dataset should maybe be simplified, in order not to make strong statements that are hard to justify.

**Documentation:**

The documentation is good, but I have some ethical concerns on the inclusion of Twitter IDs in this dataset.

**Ethics:**

Could the authors elaborate on why, in their opinion, having NSFW images in the dataset is an ethical concern? If the dataset is catching the utilization of generative models by users, having also NSFW is, in my opinion, normal. If you were to censor these images, you would end up in losing some relevant information on how people utilize generative models (therefore, the dataset would not be ecologically as valid as it is now). On the same topic, the authors claim that "NSFW content is likely to be present and popular among certain subsets of followers due to the relatively low regulation in the current landscape". It would be interesting to understand how this claim is made. Are we referring to Twitter only? If so, could the authors include a reference for this claim? This is a controversial topic, and I believe it's important to be educated on it before writing strong statements.

Ethical question: aren't Twitter IDs personal information as well? GDPR defines Personal Data as follows: (Art. 4(1))

‘personal data’ means any information relating to an identified or identifiable natural person (‘data subject’); an identifiable natural person is one who can be identified, directly or indirectly, in particular by reference to an identifier such as a name, an identification number, location data, an online identifier or to one or more factors specific to the physical, physiological, genetic, mental, economic, cultural or social identity of that natural person;

Given this, I believe that an ethical review process by an institution would have been beneficial. As I am not an ethics expert and I don't feel confident enough to express a judgement on this, I will flag the paper for ethics review.

**Limitations:**

In the "limitations" paragraph, it is not clear how the authors "estimate the bias form non-AI-generated images" to be quite small. What does "estimate" mean? Maybe they want to say that they don't have ways of estimating it and, therefore, they "assume" it? Once again, this is a strong claim that is not properly justified.

**Opportunities For Improvement:**

The proposed research questions are interesting, but the methodologies to address them are a bit convoluted and audacious. It is not clear whether we can trust the results by assuming that the methodologies are appropriate for the case study. References [6, 10, 14, 18, 35, 45] are, for example, not super consistent with the case study and, in some cases, a bit too general.

Another example of the audacity of the authors is the paragraph just before the discussion. In just 7 lines they make quite a strong claim on an important issue (copyright), implying a certain level of superficiality in the analysis. I would suggest the authors to read the whole paper again with the intention of sounding less superficial and trying to edit/remove all the parts that contribute to giving such impression.

In addition, if the methodologies could be simplified or better explained, that would be beneficial for the paper as well.

Lastly, are the authors planning to leave TWIGMA as is, or is there a possibility of creating updated versions over time, including more and more images and allowing longer longitudinal studies? (this would be an added value for sure)

**Relation To Prior Work:**

Maybe paragraph 2 should be called "Related work" instead of "Relevant work"? After reading the whole section, it's clear what TWIGMA is about, but the sentence "Recognizing the gap between [...] to answer intriguing research questions" should probably be moved later in the paper. In that position, it creates a bit of confusion (e.g., which are the intriguing research questions we are referring to? They are explained below).

**Summary And Contributions:**

The authors propose a dataset of 800.000 AI-generated images, extrapolated from Twitter over a period of time of more than one year. The authors also performed analyses on the datasets to exemplify the possible usage.

---

> ### Author Response · Authors · 2023-08-15
>
> Thank you, Reviewer Araw, for your detailed reviews and numerous points which served to improve our paper. Please see below for a point-by-point response to the concerns you’ve raised and we have worked very hard to incorporate the excellent points you raised into our revised manuscript.
>
> 1. We have revisited the choice of methodologies based on your astute comment and revised the corresponding paragraphs. We have also eliminated overly broad references:
> > Quantifying the variation of a continuous distribution is a well-studied topic across many disciplines. In this paper, we primarily drew upon existing literature that assesses diversity in natural images and generated texts and used the following metrics to capture different aspects of variations.
>
> 2. Your insightful observation about our initial copyright statement's wording has been duly acknowledged. In the revised manuscript, we have removed the strongly worded sentence and have made adjustments throughout the paper to highlight TWIGMA's applicability in understanding copyright-related matters:
> > Lastly, we used cosine similarity to extract the nearest neighbors of TWIGMA images and identified pairs with a high likelihood that the neighbors from ArtBench and LAION served as potential inspirations (i.e., training data) for the text-to-image models (see Figure4(e)). The striking similarities between some of these neighbor pairs suggest the potential of leveraging a similarity-measure-based approach to more systematically identify such pairs.
>
> 3. We thank the Reviewer for their suggestion to simplify the methods; we have gone through the paper and removed unrelated references to contextualize the used methods better and we are certainly open to further modifications based on your specific recommendations to enhance clarity.
>
> 4. Thank you for providing a platform to discuss our future plans with TWIGMA. We have updated the dataset documentation to clarify the authors' responsibility for data maintenance. Additionally, we emphasize the importance of ongoing updates to enable longitudinal studies in the future work section:
> > Moreover, maintaining regular updates to the TWIGMA dataset in order to consistently monitor the trajectory of themes and contents of AI-generated images would provide valuable insights for the research community.
>
> 5. Addressing your comment on our initial imprecise statement of "estimate the bias from non-AI-generated images" to be quite small, we have now rewritten the statement to be more precise:
> > Additionally, it is possible that a subset of images within TWIGMA are non-AI-generated, as users sometimes share relevant non-AI-generated content, such as real images closely resembling AI-generated outputs or screenshots from model websites or APIs. All of the aforementioned data issues could bias our findings in Section 4.1.
>
> 6. Thank you for your feedback on the inconsistent use of verb tenses in our prior manuscript, we have gone through the manuscript and carefully modified the tenses so that only actions in the method section referring to what we *did* (e.g., used clustering, applied filter) are in the past tense and we have tried our best to keep a consistent simple present or present perfect tense for describing the results snd current work elsewhere.
>
> 7. Thank you for recommending the modification of the section title to "Related Work”. We have incorporated this change in the revision and  tweaked the wording of relevant sentences accordingly.
>
> 8. Regarding the ethical concern (or the lack thereof) of the NSFW pictures, we appreciate the opportunity to clarify that we did not censor these images but provided a tag for potentially sensitive content to enable user choice.
>
> 9. To follow up on 8., we also wanna extend our appreciation for your guidance in recommending an ethical review, and we have included the concluding remarks of the expert Reviewer below:
> > I believe this paper adequately addresses the major ethical concerns raised by the reviewers. There may be some difficulty given developments around access to the Twitter API for distribution of this dataset, but I do not believe this rises to the level that should restrict the process for the paper. [Expert ethics review]
> With that in mind, we will keep engaging with potential concerns as users and other scholars in the field raise them, as well as when access policies for Twitter/X change in the future.
>
> 10. Finally, thank you for flagging that our initial submission mistakenly used the incorrect template. In the revision, we switched to the correct template with line numbers shown on the right hand side. P.S.: The revised manuscript will be within page limit once we switch to camera ready template.

---

> > ### Comment · Reviewer_Araw · 2023-08-26
> > **Mark raised**
> >
> > Dear authors, thank you for doing your best in addressing my concerns. I have raised my mark from 4 to 7. I believe that if you do updates of the dataset, then the longitudinal case studies could be quite interesting.

---

> > > ### Author Response · Authors · 2023-08-27
> > >
> > > Thank you for your fantastic reviews and the updated rating! We will continue to monitor the exciting longitudinal evolution of generative text-to-images contents.

---

### Official Review · Reviewer_VLov · 2023-07-21
**An interesting dataset but the usage is doubtful**

**Rating:** 5
**Confidence:** 5
**Correctness:** Yes
**Clarity:** See Opportunities For Improvement

**Strengths:**

1. The process of how to construct the dataset is detailly described.
2. The paper is easy to read and follow.


**Additional Feedback:**

N.A.

**Documentation:**

Lacks a URL for reviewer access to the dataset.

**Ethics:**

No concerns

**Limitations:**

The authors have addressed the limitations.

**Opportunities For Improvement:**

1. This paper lacks a quantitative and structured comparison (e.g., in a table) with the existing datasets, such as DiffusionDB [49]. A table can contain image numbers, prompt numbers, etc.
2. The significance of this TWIGMA dataset is doubtful. Although this paper analyzes the AI-generated content in this dataset, the practical usage of this dataset is not discussed. Also, the conclusions seem straightforward to anticipate in advance. Besides the analysis conducted in the paper, what can we use this dataset for?


**Relation To Prior Work:**

Yes

**Summary And Contributions:**

This paper collects 800,000 AI-generated photos with associated metadata to build a new TWIGMA dataset. This dataset enables the analysis of AI-generated images on Twitter, especially the difference between AI-generated images and non-AI-generated content.

---

> ### Author Response · Authors · 2023-08-15
>
> We thank the Reviewer VLov for their detailed suggestion and thoughtful review of our paper; we have carefully addressed your comments and suggestions in our revision. Before we start the point-by-point response, we first note that links to our dataset can be found at https://yiqunchen.github.io/TWIGMA/.
>
> 1. We added a table in this revision that details the dataset size, source, as well as additional metadata features in *Table 1 of the revised manuscript*. We are thankful for thre opportunity to clarify that the release of TWIGMA does not undermine but in fact highlighta the contribution of large-scale datasets such as LAION-5B and DiffusionDB; our key point is that TWIGMA serves to fill the gap of relatively short time span of datasets like DiffusionDB and Kaggle dataset on midjourney. p
>
> 2. We are appreciative of the Reviewer's keen observations concerning the significance of our work. We recognize that our prior description lacked precision regarding the potential impact of curating and analyzing datasets like ours. In response, we have further clarified the broader implications of our efforts. Our research not only holds relevance from a machine learning standpoint, where we foresee the TWIGMA dataset and its future iterations catalyzing refinement variants based on the highest-liked photos and styles, but also provides insights into human interaction with generative models in a broader scientific context. As generative content continues to proliferate, our study strives to unveil insights into the output generated by data-centric AI tools, such as diffusion models. In the revised manuscript, we now state that:
>
> > In addition to offering opportunities for fine-tuned adaptations to generate images with high popularity on social platforms like Twitter, our dataset and accompanying analyses contribute substantial insights into the ways in which humans engage with generative models, a topic of much interest to a broader scientific framework: as the prevalence of generative content continues to grow, our study could serve as pilot for future endeavors aimed at comprehending and enhancing the sociological [41] and human-machine interaction [10, 30] dimensions inherent to these data-driven gen-AI models. [Section 1]
>
> > As another example, one potential follow-up study could leverage a large pool of human annotators to more comprehensively investigate two pivotal aspects: (i) the alignment between human comprehension and categorization of TWIGMA images and scalable machine learning ouputs; and (ii) the correlation between human ratings and the number of likes for an image (included as part of TWIGMA). These studies could empower the development of future generative AI models that aim to align more closely with specific human preferences and image styles. [Future work paragraph of Section 5]
>
> Thank you for your considerations of our work in advance!

---

> > ### Author Response · Authors · 2023-08-28
> >
> > Dear Reviewers,
> >
> > Thank you again for your valuable insights and dedicated time spent in enhancing our paper.
> >
> > In response to the initial comments provided, we have diligently addressed each Reviewer's feedback and suggestions individually. In particular, we have elaborated on the potential of different use cases of our datasets and included a detailed table to refer to existing datasets used in our analysis (addressing your invaluable opportunity for improvement point 1)
> >
> >  As the deadline for discussions approaches within 1 day, we would love to see whether you would have a chance to provide additional feedback, especially on whether the responses we have provided adequately address the concerns you raised.
> >
> > Once again, we sincerely appreciate your commitment to the review process, which has been invaluable in refining our work.

---

### Decision · Program_Chairs · 2023-09-22

**Decision:**

Accept (Poster)

**Comment:**

This paper presents a novel dataset on a timely topic and accompanying analyses of that dataset to analyze its contents. The authors have made relevant changes following reviewer feedback and have already significantly improved their submission.
One hesitation I have is given changes in Twitter's API, researchers no longer have the possibility to "hydrate" the dataset contents by adding the images themselves, which limits the directions of future work. Perhaps the authors should make the images available upon request, or explore other options of sharing them. I would like this to be reflected in future versions of the paper.